# Potential of Ocular Transmission of SARS-CoV-2: A Review

**Brad P. Barnett** [1],*, **Karl Wahlin** [2], **Michal Krawczyk** [2], **Doran Spencer** [2], **Derek Welsbie** [2], **Natalie Afshari** [2] **and Daniel Chao** [2]

1   NVISION Eye Centers—South Sacramento, 7501 Hospital Dr. Suite 105, Sacramento, CA 95823, USA
2   Shiley Eye Institute, Viterbi Family Department of Ophthalmology, University of California San Diego, La Jolla, CA 92093, USA; kwahlin@health.ucsd.edu (K.W.); mkrawczyk@health.ucsd.edu (M.K.); dbspencer@health.ucsd.edu (D.S.); dwelsbie@health.ucsd.edu (D.W.); naafshari@health.ucsd.edu (N.A.); d6chao@health.ucsd.edu (D.C.)
*   Correspondence: Brad.Barnett@nvisioncenters.com; Tel.: +91-6423-4040; Fax: +91-6689-2100

**Abstract:** Purpose of review: to provide a prospective on the current mechanisms by which SARS-CoV-2 enters cells and replicates, and its implications for ocular transmission. The literature was analyzed to understand ocular transmission as well as molecular mechanisms by which SARS-CoV-2 enters cells and replicates. Analysis of gene expression profiles from available datasets, published immunohistochemistry, as well as current literature was reviewed, to assess the likelihood that ocular inoculation of SARS-CoV-2 results in systemic infection. Recent findings: The ocular surface and retina have the necessary proteins, Transmembrane Serine Protease 2 (TMPRSS2), CD147, Angiotensin-Converting Enzyme 2 (ACE2) and Cathepsin L (CTSL) necessary to be infected with SARS-CoV-2. In addition to direct ocular infection, virus carried by tears through the nasolacrimal duct to nasal epithelium represent a means of ocular inoculation. Summary: There is evidence that SARS-CoV-2 may either directly infect cells on the ocular surface, or virus can be carried by tears through the nasolacrimal duct to infect the nasal or gastrointestinal epithelium.

**Keywords:** SARS-CoV-2; COVID-19; ocular transmission; conjunctivitis; coronavirus

## 1. Introduction

Due to the widespread impact of the COVID-19 pandemic, an in depth understanding of the mechanisms of SARS-CoV-2 infection on both the macro and molecular scale is needed. Due to the ever-shifting landscape of our understanding of the virus, any consensus statement must be taken with a grain of salt. Conceding this shortcoming, for the ophthalmological community we have attempted to summarize, as well as reinterpret the literature to date as it relates to this virus. In addition to summarizing the latest data as it relates to ocular transmission of SARS-CoV-2, we have also relied on historical examples to place these new and sometimes tentative reports within a framework of understanding. The molecular mechanisms of SARS-CoV-2 infection have proven distinct from other coronaviruses. For example, dipeptidyl peptidase 4 (DPP4) is an entry receptor for MERS-CoV, but is thought to be uninvolved in SARS-CoV-2 infection [1]. Despite these distinctions, fortuitously, the mechanism of transmission of SARS-CoV-2 in multiple respects parallels that of SARS-CoV-1 and MERS-CoV. This paper highlights these parallels to guide the ophthalmological community in making educated inferences on SARS-CoV-2 and how it relates, thereby, providing a roadmap to guide preventative and therapeutic interventions.

In addition to inhalational spread, evaluation of other means of transmission, such as ocular transmission may have a far-reaching impact on patient, clinician, and ultimately society as a whole.

In regards to ocular transmission of SARS-CoV-2, three principal questions arise. 1. Does ocular inoculation of virus result in systemic infection? 2. Do ocular secretions contain live virus? 3. What precautions or treatments should be introduced to mitigate questions 1 or 2?

Much of our current knowledge of SARS-CoV-2 transmission is inferred from SARS-CoV-1 transmission. Fortuitously, the molecular similarity of SARS-CoV-1 and SARS-CoV-2 has enabled rapid diagnostic and therapeutic advances in a very short time. Moreover, the striking similarities of these viruses and the pathways of infection lend credence to the validity of these inferences. Unlike SARS-CoV-2 in which many infected persons are relatively asymptomatic, SARS-CoV-1 uniformly results in a severe viral pneumonia and in many cases death [1]. SARS-CoV-1 has been shown to be transmitted through direct contact or with droplet or aerosolized particle contact with the mucous membranes of the eyes, nose and mouth [1]. Indeed, during the 2003 SARS-CoV-1 outbreak in Toronto, health care workers who failed to wear eye protection in caring for patients infected with SARS-CoV-1 had a higher rate of seroconversion [2].

## 2. Methods

In order to provide a relative comprehensive summary of what is known about SARS-COV-2 as it relates to ocular transmission and symptoms, a search of variety of databases dedicated to compiling COVID-19 research was performed. Databases utilized for our search included MEDLINE COVID-19 database (https://www.medline.com/covid19), NIH COVID-19 database (https://www.nih.gov/health-information/coronavirus), CDC COVID-19 website (coronavirus.gov), COVID-19 articles in PubMed with the search term "((wuhan[All Fields] AND ("coronavirus"[MeSH Terms] OR "coronavirus" [All Fields])) AND 2019/12[PDAT]: 2030[PDAT]) OR 2019-nCoV[All Fields] OR 2019nCoV[All Fields] OR COVID-19[All Fields] OR SARS-CoV-2[All Fields]", CORD-19 COVID-19 Open Research Dataset from the Allen Institute (https://www.semanticscholar.org/cord19).

We identified a total of 60 articles that are relevant to ocular transmission of SARS-CoV-2. From this literature search, four key proteins implicated in SARS-CoV-2 infection were identified: Transmembrane Serine Protease 2 (TMPRSS2), CD147, Angiotensin-Converting Enzyme 2 (ACE2) and Cathepsin L (CTSL), and datasets were utilized to analyze the expression of these proteins in ocular and non-ocular tissue. From the identified articles, immunohistochemistry images for TMPRSS2, CD147, ACE2 and CTSL in ocular tissues were also analyzed.

## 3. What Are the Molecular Mechanisms by Which the SARS-Cov-2 Virus Infect Cells?

The respiratory epithelium, an established site of SARS-CoV-2 infection [3], and the epithelium of the ocular surface share many similarities. The similarities in these tissues, including the presence of goblet cells and the expression of p63, K3 and MUC5AC, have led groups to explore nasal epithelial grafts to the ocular surface to treat cicatricial disease [4]. Proteins involved in the entry of SARS-CoV-2 into cells as well as its replication have been identified, which are the targets of many therapeutics being developed. Based upon our current understanding, to best evaluate which tissues are susceptible to ocular infection, a few key proteins should be evaluated; namely TMPRSS2, CD147, ACE-2 and CTSL. The implications of how these proteins relate to virus cell fusion are summarized in Figure 1. It is important to note that this proposed pathway of infection is in no way canonical. For example, recent evidence suggests that Angiotensin II Receptor type 2 (AGTR) is a putative receptor for the virus [5].

TMPRSS2 is a serine protease that has been shown to activate viral spike glycoproteins through proteolytic cleavage. When viral spike proteins are synthesized, they are maintained in a precursor folded state incapable of interacting with host cell receptors. Once cleaved by TMPRSS2, the spike glycoprotein can interact with a cell surface receptor, either CD147 or ACE-2, thereby facilitating virus-cell fusion. CD147, also called Basigin or EMMPRIN, is a type 1 transmembrane protein that belongs to the immunoglobulin superfamily. CD147 is utilized by the host for nutrient transport, leukocyte migration and matrix metalloproteinase expression. It is also utilized by multiple human

pathogens as a cellular receptor [6]. ACE2 is a transmembrane dipeptidyl carboxydipeptidases that acts as a functional receptor for the spike glycoprotein [6]. CTSL is a lysosomal endopeptidase of the C1 family that is known to be involved in protein degradation as well as activation of certain endocytosed viruses [7].

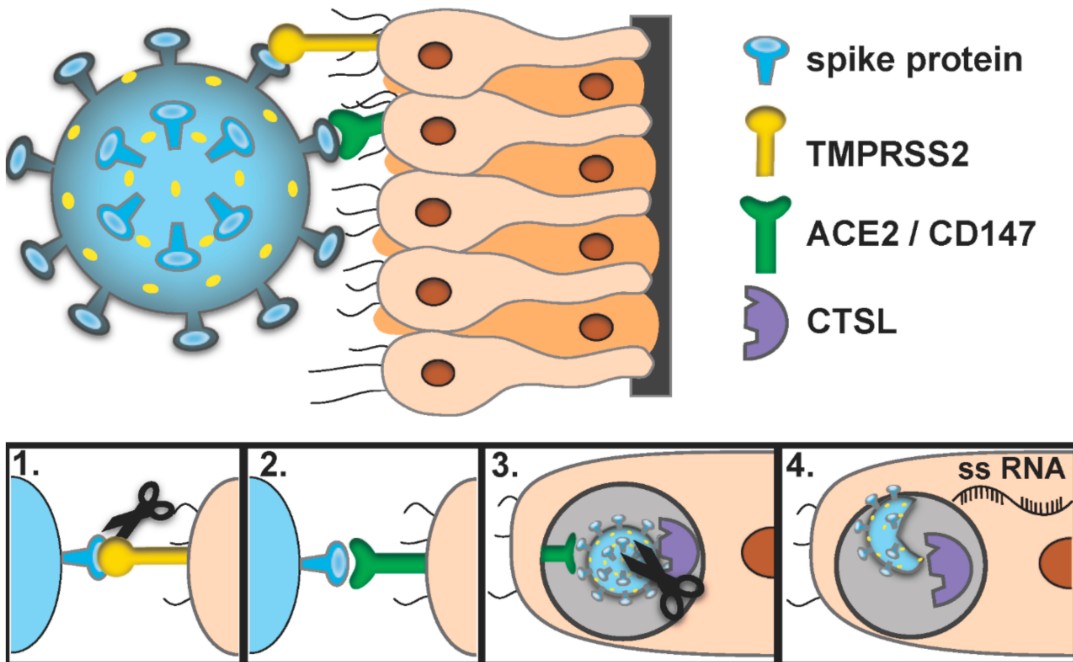

**Figure 1.** SARS-COV-2 infection pathway. (**1**). SARS-CoV-2 has a spike glycoprotein that must first be cleaved by TMPRSS2. (**2**). Spike protein interacts with CD147 or Angiotensin Converting Enzyme 2 (ACE-2). (**3**). The virus is endocytosed and cathepsin L (CTSL) must further cleave the virus. (**4**). The virus fuses with the endosome membrane, thereby releasing single stranded RNA (ssRNA) into the cytosol.

Both SARS-CoV-1 and SARS-CoV-2 have a spike glycoprotein that interacts with ACE-2 [8]. This interaction is believed to be necessary for viral penetration of the host cell [8]. There is also some evidence that the transmembrane receptor CD147 may also act as a viral receptor in lieu of ACE-2 [6]. Prior to interaction with either receptor, the spike protein must first be cleaved by the protease TMPRSS2 [9]. After this cleavage event, the virus binds to the receptor and is endocytosed [9]. It is unclear if TMPRSS2 and ACE-2 or CD147 must be on the same cell. Taking the ocular surface for example, if TMPRSS2 was independently expressed on one cell population and the receptors on another cell type, infection could still be theoretically possible as the TMPRSS2 primed virus in the tear film could be carried to the receptor laden cell population.

Once endocytosed, the virus is trapped within an endosome. It is believed that cathepsin L (CTSL) must further cleave the virus to enable it to fuse with the endosome membrane, thereby releasing single stranded RNA (ssRNA) into the cytosol (Figure 1) [10]. Although, TMPRSS2 is not requisitely expressed on a cell with receptor ligand (i.e., CD147 or ACE-2), it stands to reason that cells susceptible to SARS-CoV-2 infection must coexpress either CD147 and CTSL or ACE-2 and CTSL [9,11–13]. An intriguing study looking at the relative expression of CD147 and CTSL, demonstrated increased expression with increasing age and male sex [14].

In regards to direct ocular infection, CD147 has been critically shown to be expressed in the retina as well as the ocular surface [15]. CD147 has been shown to be upregulated in ocular surface disease [16,17]. For this reason, the use of silicone punctal plugs or intracanalicular plugs may dually block transport of

virus from the ocular surface to the highly susceptible nasal epithelium, but could also act to optimize the ocular surface thereby reducing relative expression of CD147. The interplay of CD147 and the ocular surface also highlights the potential double-edged sword of using agents such as HypoChlor (OCuSOFT; Rosenberg, TX, USA) to kill virus. This agent, which contains 0.02% hypochlorous acid and is used externally on the lids to treat blepharitis, has the ability to kill SARS-CoV-2 [18] but if directly applied to the cornea and conjunctiva instead of used as intended, externally on the lids, will certainly worsen ocular surface disease and thus expression of the CD147 receptor.

It remains unclear if SARS-CoV-2 results in retinal manifestations [19]. It is notable that ACE-2 is felt to be absent from the retina but CD147 is present [12,15,16,20–27]. As presence of CD147 or ACE2 should be sufficient for infection, the presence of CD147 in the retina suggests SARS-CoV-2 could directly infect the retina [12,15]. Moreover, CTSL is known to be expressed in the retina [28,29]. The presence of CD147 and CTSL in retina may explain why animals with coronavirus infection have retinal manifestations [28,29].

## 4. Can the Eye Serve as a Gateway for SARS-CoV-2 Transmission?

Multiple animal models have shown that transmission of coronaviruses can occur via ocular inoculation [30]. Ferrets can develop conjunctivitis after ocular inoculation with SARS-CoV-1 [31]. Numerous other species including mice, rats, cats and pigs have also developed systemic coronavirus infection after ocular inoculation [29,32–34]. Interestingly, even when oronasal inoculation was employed instead of ocular inoculation, conjunctivitis did occur with active infection [29,32–34]. It should be noted that similar experiments with SARS-CoV-2 have not yet been reported. Taken together, although conjunctivitis is not uniformly present with coronavirus infection, ocular exposure may pose a risk for transmission. Many unanswered questions remain. For example in the report of 0.8% of patients with conjunctivitis among patients with COVID-19 [35], it is conceivable that none of these patients acquired the infection through the direct ocular inoculation. It is also conceivable, in the setting of high levels of asymptomatic infection, that the number of persons with conjunctivitis far under-represents the number of individuals that are infected through ocular inoculation.

There are two pathways by which ocular exposure could lead to systemic transmission of the SARS-CoV-2 virus. (1) Direct infection of ocular tissues including cornea, conjunctiva, lacrimal gland, meibomian glands from virus exposure and (2) virus in the tears, which then goes through the nasolacrimal duct to infect the nasal or gastrointestinal epithelium (Figure 2). There is evidence now that SARS-CoV-2 virus can infect conjunctiva in vitro, but how often this occurs in patients and whether this can lead to transmission of the virus systemically is still unknown. A case report of a health care worker, wearing only a respirator, who developed conjunctivitis several days prior to developing pneumonia from SARS-CoV-2 infection, is provocative but far from confirmatory [36].

To investigate the possibility of ocular infection, we looked at the published literature to see if key components necessary for SARS-CoV-2 viral entry and replication are found in ocular tissues. We analyzed previously published gene expression data sets as well as published immunohistochemistry focusing on proteins thought to be necessary for SARS-CoV-2 infection. TMPRSS2, CD147, ACE-2 and CTSL were all expressed in ocular and non-ocular tissues that are recognized as targets for SARS-CoV-2 infection (Figure 3) [12,15,16,20–27,37–40]. Expression of these proteins in ocular tissues suggests that direct infection of ocular tissue is a possibility (Figures 3 and 4) [12,15,16,20–27].

Future experiments will need to test this hypothesis more directly. A review of the literature [41–52] and a recent meta-analysis reported in *Lancet* [53] demonstrates that lack of ocular protection increases the risk of contracting MERS, SARS-CoV-1 and SARS-CoV-2 (Figure 5). This result is further supported by a recent JAMA article that demonstrated the use of a face shield reduced seroconversion in community health workers from 19% to 0% [54]. Face shield use, unlike direct eye protection such as goggles, complicates assessment of direct ocular transmission. In addition to ocular surface protection, face shields could also act to reduce respiratory or gastrointestinal exposure. Importantly, even the

protective effect of goggles does not imply virus directly invades the ocular surface as virus can be carried via tear drainage into the nasal or gastrointestinal epithelium where infection can occur.

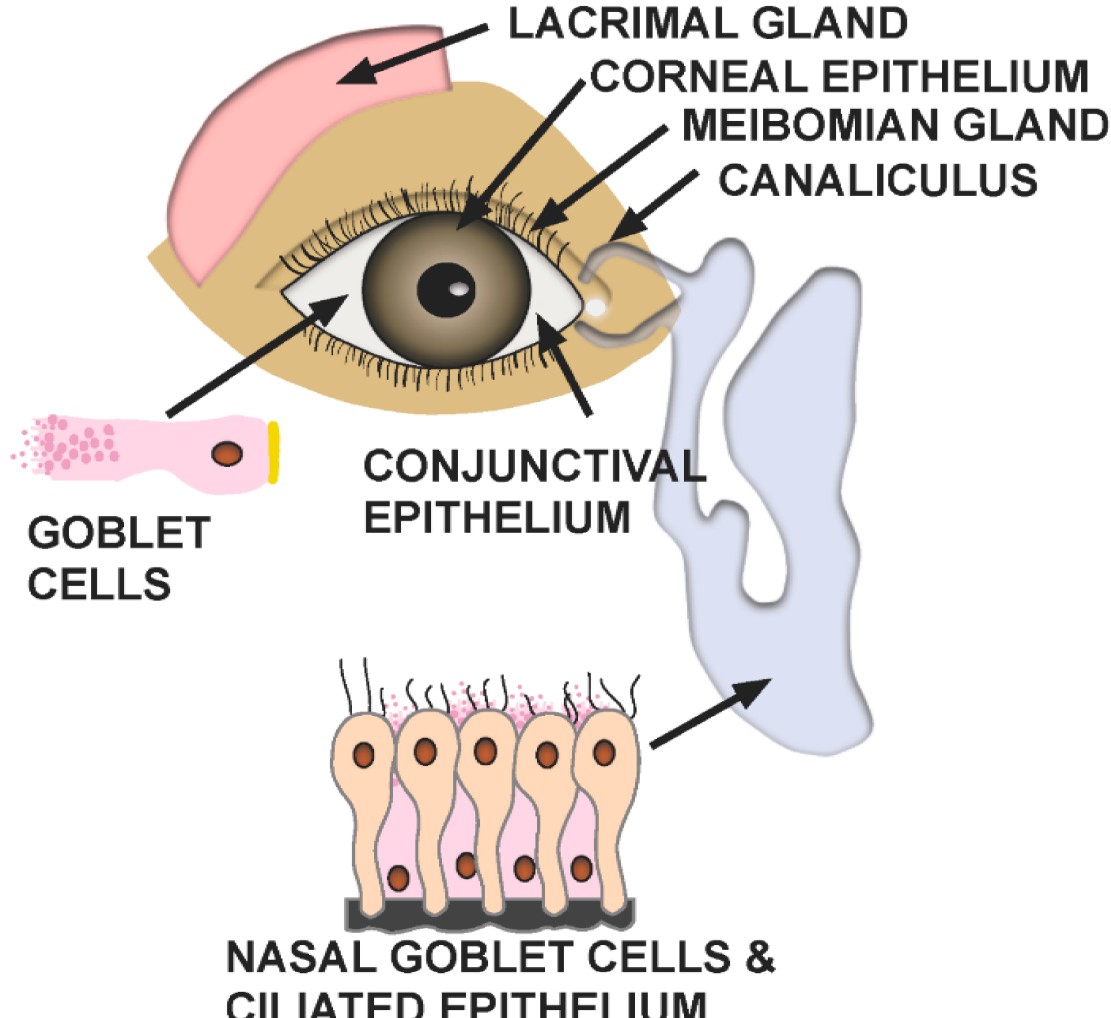

**Figure 2.** Structures at risk of infection from ocular exposure of SARS-CoV-2. Two potential pathways by which ocular exposure could lead to transmission of the SARS-CoV-2 virus include direct infection of ocular tissues as well as infection of nasal or gastrointestinal epithelium due to virus carried by tears through the nasolacrimal duct.

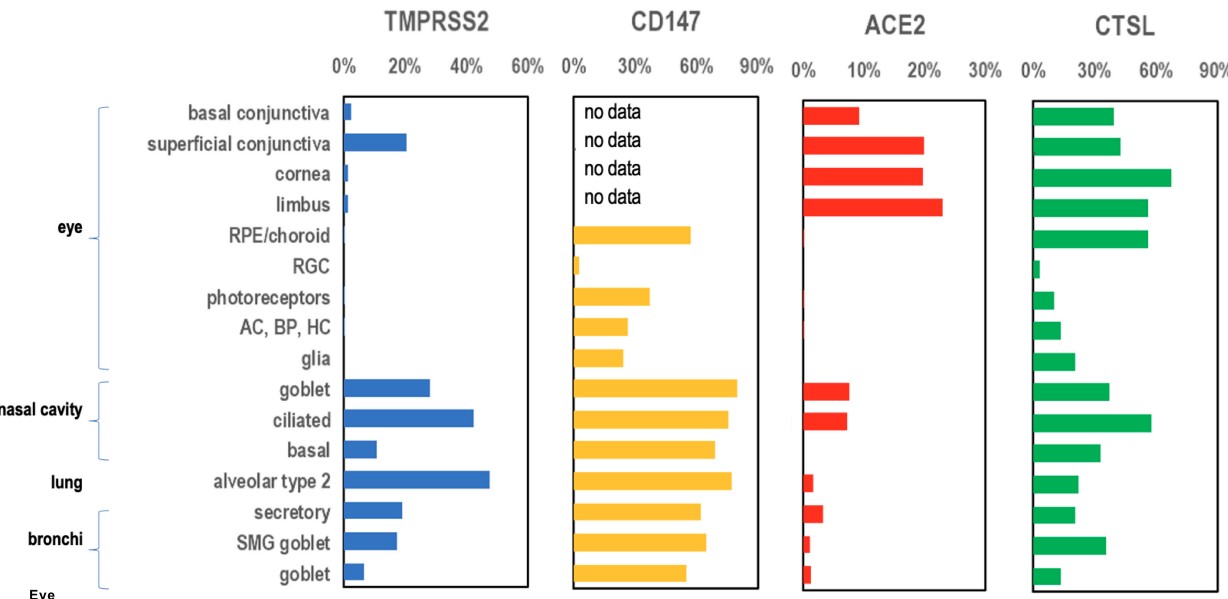

**Eye**
- Collin, J.; Queen, R.; Zerti, D.; Dorgau, B.; Georgiou, M.; Djidrovski, I.; Hussain, R.; Coxhead, J.M.; Joseph, A.; Rooney, P., et al. Co-expression of SARS-CoV-2 entry genes in the superficial adult human conjunctival, limbal and corneal epithelium suggests an additional route of entry via the ocular surface. Ocul Surf 2020, 10.1016/j.jtos.2020.05.013, doi:10.1016/j.jtos.2020.05.013.
- Menon M, Mohammadi S, Davila-Velderrain J, et al. Single-cell transcriptomic atlas of the human retina identifies cell types associated with age-related macular degeneration. Nat Commun. 2019;10(1):4902. Published 2019 Oct 25. doi:10.1038/s41467-019-12780-8
- Voigt AP, Mulfaul K, Mullin NK, et al. Single-cell transcriptomics of the human retinal pigment epithelium and choroid in health and macular degeneration. Proc Natl Acad Sci U S A. 2019;116(48):24100-24107. doi:10.1073/pnas.1914143116
- Lukowski SW, Lo CY, Sharov AA, et al. A single-cell transcriptome atlas of the adult human retina. EMBO J. 2019;38(18):e100811. doi:10.15252/embj.2018100811

**Nasal**
- Vieira Braga FA, Kar G, Berg M, et al. A cellular census of human lungs identifies novel cell states in health and in asthma. *Nat Med*. 2019;25(7):1153-1163. doi:10.1038/s41591-019-0468-5

**Lung**
- Vieira Braga FA, Kar G, Berg M, et al. A cellular census of human lungs identifies novel cell states in health and in asthma. Nat Med. 2019;25(7):1153-1163. doi:10.1038/s41591-019-0468-5
- Madissoon E, Wilbrey-Clark A, Miragaia RJ, et al. scRNA-seq assessment of the human lung, spleen, and esophagus tissue stability after cold preservation. Genome Biol. 2019;21(1):1. Published 2019 Dec 31. doi:10.1186/s13059-019-1906-x

**Bronchi**
- Soeren Lukassen, Robert Lorenz Chua, Timo Trefzer, Nicolas C Kahn, Marc A Schneider, Thomas Muley, Hauke Winter, Michael Meister, Carmen Veith, Agnes W Boots, Bianca P Hennig, Michael Kreuter, Christian Conrad, Roland Eils. SARS-CoV-2 receptor ACE2 and TMPRSS2 are primarily expressed in bronchial transient secretory cells. EMBO J (2020)39:e105114https://doi.org/10.15252/embj.20105114
- Marie Deprez, Laure-Emmanuelle Zaragosi, Marin Truchi, Sandra Ruiz Garcia, Marie-Jeanne Arguel, Kevin Lebrigand, Agnès Paquet, Dana Pee'r, Charles-Hugo Marquette, Sylvie Leroy, Pascal Barbry. A single-cell atlas of the human healthy airways, bioRxiv 2019.12.21.884759; doi: https://doi.org/10.1101/2019.12.21.884759

**Figure 3.** Gene expression of TMPRSS2, CD147, ACE-2 and CTSL. Graph of relative global gene expression in ocular and non-ocular tissue. TMPRSS2, CD147, ACE-2 and CTSL are all expressed in ocular tissues and non-ocular tissues that are currently accepted as sites for SARS-CoV-2 infection.

| | basal conjunctiva | superficial conjunctiva | cornea | limbus | lacrimal gland | RPE / choroid | RGC | photo-receptors | AC, BP, HC | glia |
|---|---|---|---|---|---|---|---|---|---|---|
| ACE2 | + | **+** | + | + | - | - | - | - | - | - |
| TMPRSS2 | + | **+** | + | + | no data | - | - | - | - | - |
| CTSL | **+** | **+** | **+** | **+** | no data | **+** | + | **+** | **+** | **+** |
| CD147 | **+** | **+** | **+** | no data | + | **+** | + | **+** | **+** | **+** |

- Aakalu VK, Parameswaran S, Maienschein-Cline M, Bahroos N et al. Human Lacrimal Gland Gene Expression. PLoS 2017;12(1):e0169346.
- Huet, E., Vallée, B., Delbé, J., Mourah, S., Prulire-Escabasse, V., Tremouilleres, M., … Gabison, E. E. (2011). EMMPRIN modulates epithelial barrier function through a MMP mediated occludin cleavage: Implications in dry eye disease. American Journal of Pathology, 179(3), 1278–1286. https://doi.org/10.1016/j.ajpath.2011.05.036
- Kitazawa K, Hikichi T, Nakamura T, Mitsunaga K et al. OVOL2 Maintains the Transcriptional Program of Human Corneal Epithelium by Suppressing Epithelial-to-Mesenchymal Transition. Cell Rep 2016 May 10;15(6):1359-68
- Leonardi A, Rosani U, Brun P. Ocular expression of SARS-CoV-2 Receptors. Ocular Immunology and Inflammation, 28:5, 735-738, DOI:
- 10.1080/09273948.2020.1772314
- Lukowski SW, Lo CY, Sharov AA, et al. A single-cell transcriptome atlas of the adult human retina. EMBO J. 2019;38(18):e100811. doi:10.15252/embj.2018100811
- Määttä M, Tervahartiala T, Kaarniranta K, et al. Immunolocalization of EMMPRIN (CD147) in the human eye and detection of soluble form of EMMPRIN in ocular fluids. Curr Eye Res. 2006;31(11):917-924. doi:10.1080/02713680600932290
- Menon M, Mohammadi S, Davila-Velderrain J, et al. Single-cell transcriptomic atlas of the human retina identifies cell types associated with age-related macular degeneration. Nat Commun. 2019;10(1):4902. Published 2019 Oct 25. doi:10.1038/s41467-019-12780-8
- Collin, J.; Queen, R.; Zerti, D.; Dorgau, B.; Georgiou, M.; Djidrovski, I.; Hussain, R.; Coxhead, J.M.; Joseph, A.; Rooney, P., et al. Co-expression of SARS-CoV-2 entry genes in the superficial adult human conjunctival, limbal and corneal epithelium suggests an additional route of entry via the ocular surface. Ocul Surf 2020, 10.1016/j.jtos.2020.05.013, doi:10.1016/j.jtos.2020.05.013.
- Tong L, Diebold Y, Calonge M, Gao J et al. Comparison of gene expression profiles of conjunctival cell lines with primary cultured conjunctival epithelial cells and human conjunctival tissue. Gene Expr 2009;14(5):265-78.
- Turner HC, Budak MT, Akinci MA, Wolosin JM. Comparative analysis of human conjunctival and corneal epithelial gene expression with oligonucleotide microarrays. Invest Ophthalmol Vis Sci 2007 May;48(5):2050-61.
- Voigt AP, Mulfaul K, Mullin NK, et al. Single-cell transcriptomics of the human retinal pigment epithelium and choroid in health and macular degeneration. Proc Natl Acad Sci U S A. 2019;116(48):24100-24107. doi:10.1073/pnas.1914143116

**Figure 4.** Relative expression of TMPRSS2, CD147, ACE-2 and CTSL in ocular structures. The figure summarizes data available from immunohistochemistry and gene expression profiles. (-) No expression, (+) Low expression, (+) Moderate expression, (**+**) High Expression.

| | Country | Respirator (0=no) | Events, eye protection (n/N) | Events, no eye protection (n/N) | | RR (95% CI) | % weight (random) |
|---|---|---|---|---|---|---|---|
| **MERS** | | | | | | | |
| Alraddadi et al (2016)[34] | Saudi Arabia | 1 | 1/47 | 17/165 | | 0·21 (0·03–1·51) | 4·0 |
| Ki et al (2019)[47] | South Korea | 1 | 0/9 | 6/64 | | 0·50 (0·03–8·21) | 2·2 |
| Kim et al (2016)[49] | South Korea | 1 | 0/443 | 2/294 | | 0·13 (0·01–2·76) | 1·8 |
| Ryu et al (2019)[65] | South Korea | 1 | 0/24 | 0/10 | | (Not calculable) | 0 |
| Random subtotal ($I^2$=0%) | | | 1/523 | 25/533 | | 0·24 (0·06–0·99) | 8·0 |
| | | | | | | | |
| **SARS** | | | | | | | |
| Chen et al (2009)[39] | China | 0 | 1/45 | 90/703 | | 0·17 (0·02–1·22) | 4·2 |
| Liu et al (2009)[51] | China | 0 | 17/221 | 34/256 | | 0·58 (0·33–1·01) | 21·2 |
| Pei et al (2006)[61] | China | 0 | 24/120 | 123/323 | | 0·53 (0·36–0·77) | 26·0 |
| Yin et al (2004)[75] | China | 0 | 10/120 | 67/137 | | 0·17 (0·09–0·32) | 19·4 |
| Caputo et al (2006)[38] | Canada | 1 | 2/46 | 4/32 | | 0·35 (0·07–1·79) | 5·6 |
| Ma et al (2004)[54] | China | 1 | 7/175 | 40/269 | | 0·27 (0·12–0·59) | 15·6 |
| Park et al (2004)[58] | USA | 1 | 0/30 | 0/72 | | (Not calculable) | 0 |
| Peck et al (2004)[60] | USA | 1 | 0/13 | 0/19 | | (Not calculable) | 0 |
| Random subtotal ($I^2$=62%) | | | 61/770 | 358/1811 | | 0·34 (0·21–0·56) | 92·0 |
| | | | | | | | |
| **COVID-19** | | | | | | | |
| Burke et al (2020)[37] | USA | 1 | 0/42 | 0/34 | | (Not calculable) | 0 |
| Random subtotal | | | 0/42 | 0/34 | | (Not calculable) | 0 |
| | | | | | | | |
| Random overall ($I^2$=43%) | | | 62/1335 | 383/2378 | | 0·34 (0·22–0·52) | 100·0 |
| Adjusted estimates, overall (2 studies, Yin[75] and Ma[54]) | | | | | | aOR 0·22 (0·12–0·39) | |
| | | | | | | aRR 0·25 (0·14–0·43) | |
| Interaction by virus, p=0·75 | | | | | | | |

0·1    0·5 1 2    10

Favours eye protection    Favours no eye protection

**Figure 5.** Forest plot showing the association of eye protection with risk of MERS, SARS-CoV-1 (SARS) and SARS-CoV-2 (COVID-19). The plot diagram represents unadjusted estimates. MERS = Middle East respiratory syndrome. SARS = severe acute respiratory syndrome. RR = relative risk. aOR = adjusted odds ratio. aRR = adjusted relative risk [53].

## 5. Discussion

When evaluating the relationship between SARS-CoV-2 and the ocular surface, the presence of conjunctivitis in a subset of patients infected with SARS-CoV-2, along with detection of viral RNA in tears, supports the notion that the eye is a potentially important route for viral entry. When interpreting the data to date, it is important to remember that conjunctivitis does not imply active infection and the detection of RNA on the ocular surface does not necessarily imply a live virus is present in ocular secretions. Indeed, many patients receiving ventilation in the ICU may develop conjunctivitis purely from exposure keratopathy or other noninfectious causes. Moreover, viral RNA can be seeded on the ocular surface from exhaled air or transferring of respiratory secretions from hand to eye. Although it is generally assumed that ocular cells can be infected by SARS-CoV-2, experimental validation that conjunctival tissues can be directly infected is still somewhat limited [55]. RNA sequencing efforts to compare SARS-CoV-2 receptor expression in ocular surfaces (e.g., basal conjunctiva, superficial conjunctiva, conjunctival epithelium, corneal stroma, cornea, lacrimal gland, limbal superficial and limbal stroma) have so far confirmed that ACE2 and TMPRSS2 is expressed in conjunctival tissues and not lacrimal gland, corneal stroma or limbal stroma [3]. While this is a useful starting point, this type of analysis does not provide cell type specificity, thus it remains to be determined what the actual route of entry is.

The conjunctival epithelium and conducting airways appear to be potential important routes of entry for SARS-CoV-2 as has been demonstrated by ex-vivo human lung, conducting airway, and ocular conjunctiva explant cultures, which have been used to investigate the tropism of epidemic viruses including the 2009 pandemic influenza H1N1 (H1N1pdm) virus and MERS-CoV and more recently SARS-CoV-2 [56,57]. In the later study, SARS-CoV-2 was shown to infect ciliated, mucus-secreting, and club cells of bronchial epithelium; type 1 pneumocytes in the lung; and the conjunctival mucosa [56,57].

These cell culture experiments demonstrate fairly convincing evidence that conjunctiva can be infected, however, it remains unknown which specific cells in the conjunctival tissues are susceptible to COVID infection. The high levels of expression of ACE2 and TMPRSS2 in mucin secreting goblet cells of the nasal passages and their susceptibility to viral infection, suggests that similar goblet cells that also exist in the conjunctiva might also be susceptible to infection [3]. On the other hand, at least one recent paper suggests that ACE2 and TMPRSS2 are not even expressed in ocular goblet cells [11]. In that study, surgical conjunctival specimens showed detectable expression of ACE2 in the conjunctival and corneal epithelium, with the highest levels in the superficial epithelial cells. In the conjunctiva, TMPRSS2 was expressed throughout the whole epithelial layer, in contrast to ACE2, which was found more apically. Ocular goblet cells were essentially negative for both ACE2 and TMPRSS2 contrasting with goblet cells of the airways, which are positive [3]. Independent verification of viral susceptibility will need to be validated by viral challenge of these ocular tissues to clarify which cells (if any) are the true target of SARS-CoV-2.

Several experimental approaches could help to clarify which ocular cells are susceptible to SARS-CoV-2. First, efforts should be made to more comprehensively assess receptor levels. This can be accomplished by tissue punches using double labeling immunohistochemistry with markers for ocular cells (e.g., goblet cells) and ACES2, TMPRS2 and CD147 receptor. Next generation single (NGS) cell RNA sequencing focusing on conjunctival cell populations could also offer a level of resolution not previously observed. Impression cytology to collect conjunctival cells could provide a fast-convenient approach for assessing in the various cell populations.

Other areas of active research are stem cell derived organoid approaches that can be tailored to create a variety of complex biologically relevant 3 dimensional human tissues including vascular, lung, kidney and cornea organoids that can be selectively infected in vitro to explore cell type specific viral susceptibility [58–60]. The utility of such models was recently demonstrated with engineered human blood vessel organoids and human kidney organoids that were used to test a novel human recombinant soluble ACE2 (hrsACE2), which was shown to significantly block early stages of SARS-CoV-2 infections.

Similar efforts to engineer more complex ocular tissues are underway and could be useful in testing similar antiviral approaches.

While the organoid approach is clearly a valuable approach, it is important to note that in vitro models may not fully recapitulate in vivo biology. The innate immune system may play a role in viral load and pathogenesis. For instance, SARS-CoV-2 appears to replicate more extensively in the bronchus than SARS-CoV-1, and patients with COVID-19 admitted to the ICU typically had higher plasma concentrations of proinflammatory cytokines. Thus, there is a potentially important role of innate host responses in pathogenesis of COVID-19 and should be considered. Ex-vivo human primary tissue explants, including human conjunctiva, contain immune cells, and their use would involve innate immune responses.

If we consider ocular tissue as an entryway for transmission of the virus, one can consider various methods to prevent ocular exposure. Obviously, eye protection using safety glasses would be effective. Other approaches could be the use of topical eye drops to prevent viral spread or kill the virus. A variety of clinically approved and preclinical drugs that may be effective in reducing the rate of infection and increasing clearance of the virus that have been explored for systemic administration, may have benefit as a topically administered eye drops.

Potential medications specifically targeting TMPRSS2, CD147, ACE-2 and CTSL are also available. For example, a clinically approved CD147-monoclonal antibody, Meplazumab could potentially block viral entry. A recent small molecule CD147 inhibitor was reported that reduces corneal fibrosis in a rat model of alkali burn [17]. Azithromycin, approved use in the US and global markets as AzaSite may act to block CD147 [6]. In the same way, soluble ACE-2 receptor and antibody to the Spike protein could act to neutralize virus cell fusion.

Therapeutics that prevent TMPRSS2 or CTSL from cleaving the virus could also be a means of preventing infections [9,11]. A few of these medications are currently used in topical ocular formulations. Teicoplanin, which is used as a topical antibiotic in non-US markets, is a known CTSL inhibitor, which may as well block SARS-CoV-2 infection. A few examples of potential topical agents in the pipeline as well as clinically approved agents are featured in Figure 6. A first step would be to directly test the efficacy of these topical medications on SARS-CoV-2 infection in vitro.

| NAME | TARGET | CLASS | STRUCTURE | PHASE |
|---|---|---|---|---|
| Anti-Spike Protein | Spike Protein | Antibody | | Interventional Clinical Trial |
| Camostat Mesylate | TMPRSS2 | Small Molecule | | Clinically Approved (Outside US) |
| Anti-ACE2 | ACE2 | Antibody | | Interventional Clinical Trial |
| N-(2-aminoethyl)-1 aziridine-ethanamine | ACE2 | Small Molecule | | Pre-Clinical |
| Soluble ACE2 | ACE2 | Recombinant Protein | | Interventional Clinical Trial |
| Anti-CD147 | CD147 | Antibody | | Pre-Clinical |
| Azithromycin | CD147 | Antibiotic | | Clinically Approved |
| Z-Val-Phe-CHO | CTSL | Peptide Inhibitor | | Pre-Clinical |
| Teicoplanin | CTSL | Antibiotic | | Clinically Approved (Outside US) |

**Figure 6.** Potential topical ocular medications targeting specifically TMPRSS2, ACE-2, CD147 and CTSL.



## 6. Conclusions

Understanding methods of transmission of SARS-CoV-2 is critical to controlling the pandemic and informing public health strategy. Our analysis of the literature as well as analysis of genes involved in viral infection in ocular tissues, suggest that both direct infection of the ocular surface or transmission of the virus through tears down the nasolacrimal duct to infect the nasal epithelium are both plausible. Recent data in the laboratory demonstrates that conjunctival explants can be infected, thus ocular transmission is quite likely. Further research is needed to definitively answer these questions.

Key Points:

–   The key proteins necessary for SARS-CoV-2 infection include Transmembrane Serine Protease 2 (TMPRSS2), CD147, Angiotensin-Converting Enzyme 2 (ACE2) and Cathepsin L (CTSL).
–   The ocular surface and retina have variable expression of TMPRSS2, CD147, ACE2 and CTSL and appear to be susceptible to SARS-CoV-2 infection.
–   Nasal epithelium is known to highly express TMPRSS2, CD147, ACE2 and CTSL.
–   In addition to direct ocular infection, virus carried by tears through the nasolacrimal duct to nasal epithelium may represent a means of ocular inoculation.
–   Efforts to shield the ocular surface and prevent drainage of virus from tears into nasal epithelium may prevent SARS-CoV-2 infection.

(See Supplementary Materials for Summary Video of Key Points).

**Supplementary Materials:** The following are available online at http://www.mdpi.com/2411-5150/4/3/40/s1.

**Funding:** The work was supported by an unrestricted grant from Research to Prevent Blindness, New York, NY. D.L.C was supported by NIH/NEI K08EY030510.

**Conflicts of Interest:** The authors do not have any relevant conflicts of interest.

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
