# Peer review of "Potential of Ocular Transmission of SARS-CoV-2: A Review"

_2411-5150, 2020_

Round 1

Reviewer 1 Report

The manuscript “Potential of Ocular Transmission of SARS-CoV-2: A Review” is well written and focused on the objective. There is relevant summary of the research findings on the possible observations pointing towards the ocular surface being a possible route of infection. The authors have missed to cite some relevant literature which formed the initial evidence to point on ocular mode of transmission.

  1. Figure 2, the authors mention that there is no data on CD147 expression level in basal conjunctiva and superficial conjunctiva. The authors should refer to Leonardi et al., 2020 and update the information.
  2. The authors should consider citing Cheng-wei et al., 2020 as a first case of possible contraction of SARS-CoV2 via direct exposure to the eyes.
  3. The authors should consider stating a few facts on CD147 or mentioning the other names of the molecules (basigin).
  4. The authors have not mentioned about another probable candidate for entry of SARS CoV2 – Angiotensin II receptor type 2 (ATGR2).

Reviewer 2 Report

This is an interesting, timely, and well written manuscript with important implications. Below are some comments.

Authors mentioned that virus carried by tears carried through the nasolacrimal duct to nasal epithelium represent a means of ocular inoculation. However, no virions were detected in the tear samples from seventeen patients with COVID-19 (Seah IYJ, Ophthalmology 2020, Jul;127(7):977-979.) Please explain about that.

COVID-19 is basically a respiratory disease, and There were only 0.8% of patients with ocular symptoms among patients with COVID-19 (N Engl J Med 2020;382:1708-1720), mostly presenting with conjunctivitis. According to another paper from Ophthalmology, eight patients (6.6%) showed ocular symptoms. Please explain about that.

Authors should compare the key receptor genes implicated in COVID19 infection in ocular cells with those in the respiratory cells, then should see why COVID19 less likely infects with ocular cells, and what is different between both types of cells.  That would be helpful to clarify what is the ocular transmission of COVID19 like. 

What is the entry receptor expressed in the ocular surface cells for other viruses? ANPEP is an entry receptor for HCoV-229E), ST6GAL1/ST3GAL4 are enzymes for synthesis of influenza entry receptors and DPP4 is an entry receptor for MERS-CoV. These genes were expressed in the airway.

Reviewer 3 Report

In this paper, authors reviewed the current concept on the viral transmission of SARS-CoV-2 from the literary survey and concluded that SARS-CoV-2 may infect ocular tissue through direct transmission or nasolacrimal duct route.

Overall description of this paper was appropriate and the key mechanism of viral transmission using several molecules, such as TMPRSS2, ACE2, CD147 and CTSL were discussed in details including the histological distribution of these molecules in each ocular tissues. As indicated, “Expression of these protein in ocular tissues suggest that direct infection of ocular tissue is a possibility (lines 147 - 148), their conclusion on the ocular transmission of SARS-CoV-2 is still likely, these can be accepted considering the ongoing attempts of this field conducted by those including ophthalmologists.

If possible, more therapeutic information against SARS-CoV-2 ocular infection might be meaningful for clinical ophthalmologist, because only one agent was mentioned for clinical purpose in this paper (line 117).

Round 2

Reviewer 2 Report

Authors have provided a detailed point-by-point response to comments, addressed issues raised by the reviewer.